# The Interaction of HMGB1 with the Proinflammatory TREM-1 Receptor Generates Cytotoxic Lymphocytes Active against HLA-Negative Tumor Cells

**DOI:** 10.3390/ijms25010627

**Published:** 2024-01-03

**Authors:** Daria M. Yurkina, Elena A. Romanova, Alexey V. Feoktistov, Natalia V. Soshnikova, Anna V. Tvorogova, Denis V. Yashin, Lidia P. Sashchenko

**Affiliations:** 1Institute of Gene Biology (RAS), Moscow 119334, Russia; yrkina121@gmail.com (D.M.Y.); elrom4@rambler.ru (E.A.R.); sashchenko@genebiology.ru (L.P.S.); 2Engelhardt Institute of Molecular Biology (RAS), Moscow 119334, Russia; a.feo95@mail.ru (A.V.F.); so2615nat@gmail.com (N.V.S.); 3Center for Precision Genome Editing and Genetic Technologies for Biomedicine, Institute of Gene Biology (RAS), Moscow 119334, Russia; annatvor@mail.ru

**Keywords:** HMGB1, TREM-1, cytotoxic lymphocytes, HLA-negative tumor cells, apoptosis, necroptosis, IL-2

## Abstract

High mobility group protein (HMGB1) is secreted by myeloid cells and cells of damaged tissues during inflammation, causing inflammatory reactions through various receptors, including TLR_S_ and RAGE. TREM-1 is considered to be one of the potential HMGB1 receptors. In this work, we have shown that the HMGB1 protein is able to bind to the TREM-1 receptor at high affinity both in solution and on the cell surface. This binding causes lymphocytes to release cytokines IL-2, IL-1b, IL-6, TNF and Ifny into the medium, which leads to the appearance of cytotoxic lymphocytes in PBMC capable of lysing HLA-negative tumor cells. Expanding the spectra of proinflammatory receptor ligands and understanding the mechanisms of their action is essential for the creation of new immunotherapy pathways.

## 1. Introduction

The expression of cytokine genes and the subsequent secretion of these multifunctional proteins is considered as a key stage in the development of the immune response. Cytokines are responsible for the development of inflammation and provide activation of immune cells. The overproduction of cytokines, called a “cytokine storm” causes an excessive pro-inflammatory response and leads to damage to cells and tissues and to the development of autoimmune diseases [1,2].

The expression of proinflammatory cytokine genes depends on the activation of pathogen-recognizing receptors (PARP) during their interaction with ligands. The most well-studied PARPs include the Toll-like receptor families (TLR_S_) and the trigger receptor expressed on myeloid cells (TREM-1) [3,4].

Expanding the spectrum of proinflammatory receptor ligands and understanding the mechanisms of their functional activity is essential for identifying new inhibitors of inflammatory processes and contributes to the creation of new areas of immunotherapy. TLR_S_ play a crucial role in the regulation of innate immunity. These receptors are located mainly on the surface of immune cells and affect the development of infectious and non-infectious diseases [5,6]. TLR_S_ ligands can be both pathogen-associated molecules and endogenous molecules released from damaged tissues. The interaction of ligands with receptors leads to the association of the cytoplasmic domain with the MyD88 adapter and the activation of MAP kinases and nuclear transcription factors that cause increased gene translocation, including proinflammatory cytokine genes [7]. Activation of proinflammatory cytokines leads to the development of inflammatory processes.

The innate immunity receptor TREM-1 is present on the surface of immune cells: monocytes, neutrophils, dendritic cells, NK cells and to a lesser extent on some subpopulations of T and B lymphocytes [8]. One of the main functions of TREM-1 is the induction of proinflammatory cytokine gene expression. At the same time, it is involved in the regulation of the immune response: proliferation of T cells, activation of antigen-presenting cells, antiviral protection [9]. In synergy with other proinflammatory receptors, mainly TLR_S_, it participates in the induction of overexpression of proinflammatory cytokines (“cytokine storm”), causing a strong proinflammatory immune response and severe complications in the development of inflammatory processes [10,11].

TREM-1 has an Ig-like exodomain responsible for ligand binding, short transmembranes and cytoplasmic domains. Activation of TREM-1 is carried out due to the interaction of the ligand with the extracellular domain and subsequent dimerization of the receptor. Dimerization leads to interaction with the DAP 12 adaptive molecule and the launch of a cascade of tyrosine protein kinases, followed by activation of the NFkß factor and expression of proinflammatory cytokine genes [12,13]. Dimerization of TREM-1 is accompanied by cleavage of the exodomain and dissociation from the cell surface of soluble sTREM-1 [14]. Currently, sTREM-1 is considered a biomarker of the TREM-1 pathway activation in inflammatory processes [15].

A wide range of proteins with different functional activities is described, which are also considered potential TREM-1 ligands. Extracellular RNA-binding protein (eCIRP) causes acute lung injury (ALI) in healthy mice, which is blocked by the LP17 peptide inhibiting the interaction of the TREM-1 receptor with its ligands. The FRET (Förster resonance energy transfer) method revealed a strong affinity binding of eCIRP to TREM-1. The 7-membered eCIRP M3 peptide significantly inhibited eCIRP-induced systemic inflammation [16].

Actin, which is involved in many cellular processes, has also been shown to activate the inflammatory response when interacting with TREM-1. It was also found that recombinant actin could interact with the extracellular domain of TREM-1 and enhance the inflammatory response when injected with this protein in mice. This effect was inhibited by the LP17 peptide [17].

The innate immunity protein PGLYPR1 involved in antibacterial protection is identified as the TREM-1 ligand. The interaction of PGLYPR1 in complex with peptidoglycan with the TREM-1 receptor was established. Incubation of this complex with cells carrying TREM-1 on the surface led to rapid phosphorylation of Syk (spleen tyrosine kinase) and release of inflammatory cytokine IL-6 and sTREM-1 into the conditioned medium, which indicates activation of the TREM-1 pathway [15,18].

At our institute, it was confirmed that the innate immunity protein Tag7 (PGLYPR1) is a TREM-1 ligand. Tag7 has been shown to bind to TREM-1 on the surface of monocytes and induce a TREM-1-dependent signal. It was also found that the interaction of Tag7 with TREM-1 leads not only to increased expression of proinflammatory cytokine genes, but also to the generation of subpopulations of lymphocytes with cytotoxic effect on HLA-negative tumor cells. Activation of these cytotoxic T-lymphocytes is regulated by cytokines secreted by immune cells [19].

The main heat shock protein Hsp70 is also considered a TREM-1 ligand. It has been shown that it is present in the lysates of necrotic cells and is responsible for the induction of proinflammatory cytokine production [20]. At our institute, it was found that Hsp70 binds to TREM-1 on the surface of monocytes and, like Tag7, induces the expression of proinflammatory cytokine genes and cytokine-dependent activation of cytotoxic T-lymphocytes [21].

High mobility group protein (HMGB1) is a nuclear and cytoplasmic protein that interacts with nucleosomes, histones and transcription factors. During inflammation, HMGB1 is secreted by myeloid cells and cells of damaged tissues, causing inflammatory reactions through various receptors, including TLR_S_ and RAGE. Using the methods of cross-binding and immunoprecipitation, the interaction between HMGB1 and TREM-1 was established [22]. Necrotic cell lysate contains HMGB1 and stimulates TREM-1-dependent activation of proinflammatory cytokine and chemokine genes. However, it is considered that HMGB1 needs coactivating molecules to trigger TREM-1 activation.

The aim of this work was to characterize the functional activity of HMGB1 in inflammatory processes: (1) to confirm that HMGB1 is a TREM-1 ligand; (2) to show that HMGB1 induces TREM-1-dependent activation of cytotoxic lymphocytes that kill tumor cells that have evaded immune control.

## 2. Results

### 2.1. HMGB1 Binds to sTREM-1 with High Affinity

As mentioned above, it was previously shown that HMGB1 could interact with sTREM-1. We investigated the affinity of such an interaction using the quantitative method of microscale thermophoresis. Labeled sTREM-1 was incubated with different concentrations of HMGB1. The obtained signals demonstrate a clear binding curve (Figure 1). The dissociation constant (K_d_) calculated from these results was 3.04 ± 0.2 nM, which indicates a highly specific interaction of these proteins. It can be assumed that HMGB1 and sTREM-1 can form a stable complex.

### 2.2. HMGB1 Activates the TREM-1 Pathway

Next, we tested the interaction of HMGB1 with TREM-1 exposed on the cell surface. For this purpose, HMGB1 was incubated with monocytes in the presence of a BS^3^ crosslinking agent. The cell lysate was purified by magnetic separation and specific antibodies against TREM-1. The bound material was separated and analyzed using SDS-PAGE and Western blot with specific anti-HMGB1 antibodies (Figure 2a). It can be seen that the fraction bound to antibodies against TREM-1 contains a protein complex interacting with antibodies against HMGB1.

In this regard, it can be assumed that the HMGB1–TREM-1 complex was fixed on the BS^3^ cell surface. The molecular weight of this complex is approximately 52 kDa, which corresponds to the equimolar complex (27 kDa + 26 kDa), i.e., HMGB1 predominantly binds to the TREM-1 monomer (Figure 2a, Appendix A).

Confocal microscopy was used to confirm the binding of HMGB1 to TREM-1 on the cell surface. (Figure 3, Appendix A) Soluble HMGB1 was added to the monocytes. It is possible to see the colocalization of signals from green-stained HMGB1 and red-stained TREM-1 on the surface of monocytes obtained from peripheral blood mononuclear cells (PBMC) of healthy donors, which confirms the interaction of these proteins. Three different donors and more than 400 cells in each sample were analyzed to conclude that about 50% of all visible monocytes were double-stained (Appendix A).

Next, the activation of the TREM-1 receptor under the action of HMGB1 was investigated. It is known that the dissociation of soluble sTREM-1 from the cell membrane indicates the activation of the TREM-1 pathway. The appearance of sTREM-1 in a conditioned medium of monocytes incubated with HMGB1 was investigated by enzyme immunoassay. The results are shown in Figure 2b. It can be seen that the release of sTREM-1 is dose-dependent on the concentration of HMGB1. Therefore, it can be assumed that HMGB1 is a TREM-1 ligand and induces the dissociation of sTREM-1 from the cell membrane, indicating the activation of the receptor.

### 2.3. HMGB1 Induces Cytotoxic Activity of Lymphocytes

Having made sure that HMGB1 can be considered a TREM-1 ligand, we investigated its ability to activate cytotoxic lymphocytes capable of lysing HLA-negative tumor cells.

PBMC was incubated with HMGB1 for 6 days, cytotoxic activity was tested on the 4th, 5th and 6th days of incubation with HMGB1 using HLA-negative human erythroleukemia cells of the K562 line as targets for 3 or 20 h. It can be seen that HMGB1-activated lymphocytes exhibit cytotoxic activity at various time intervals (Figure 4b). The highest cytotoxicity was observed after 6 days, and the lowest after 5 days of incubation.

HMGB1 can be considered a ligand of two proinflammatory receptors present on monocytes: TLR and TREM-1. To find out which of these receptors is involved in the induction of cytotoxicity of lymphocytes, an inhibitory analysis was performed. In Figure 4b it can be seen that HMGB1-induced cytotoxic activity was completely suppressed in the presence of a well-studied TREM-1 inhibitor, the LP17 peptide. The recently described peptide N1, a Tag7 peptide fragment blocking the interaction of Tag7 (NVQHYHMK) with monocytic TREM-1, also significantly reduced the cytotoxicity of HMGB1-activated lymphocytes. The TLR inhibitor had no inhibitory effect on HMGB1-dependent cytotoxicity. Moreover, the known TLR ligand, LPS lipopolysaccharide, did not induce cytotoxic activity as well as the control BSA, which has no affinity to the TREM-1 receptor (Figure 4b). These results show that the cytotoxicity of lymphocytes induces the interaction of HMGB1 with TREM-1.

The dependence of cytotoxic activity on the concentration of HMGB1 was dose-dependent; half of the maximum cytotoxicity (I_50_) was 1 ± 0.2 nM (Figure 4c).

The dependence of inhibition on the concentration of LP17 was also dose-dependent; half of the maximum inhibition (I_50_) was 3 ± 0.3 nM.

### 2.4. Cytokines IL-6, IL-1ß, TNF and IL-2 Are Involved in the Activation of HMGB1 Induced Cytotoxicity

In order to characterize the immune system cells needed to transmit the activation signal under the action of HMGB1, CD14+-monocytes or CD4+-T lymphocytes were removed from the PBMC mixture by magnetic separation, and the remaining PBMC were tested for cytotoxicity. In both cases, cytotoxic activity was almost completely blocked (Figure 5a). It can be assumed that monocytes are involved in the activation of cytotoxic lymphocytes.

Previously, we showed the involvement of TNF, Ifnγ and IL-2 cytokines secreted by monocytes and CD4+-T lymphocytes in the generation of cytotoxic subpopulations of effector lymphocytes [19]. In this regard, we further tested the expression of proinflammatory cytokine genes and the secretion of IL-2 cytokine. Changes in cytokine levels were analyzed using qPCR. After incubation of PBMC with HMGB1 for 3 h, an increase in the level of mRNA of proinflammatory cytokines IL-6, IL-1ß, TNF and Ifnγ was observed both in PBMC and in an isolated population of monocytes. Controls for nonspecific PBMC activation and qtPCR on isolated CD4+ T cells are presented in Appendix A. Incubation of cells with HMGB1 in the presence of an LP17 inhibitor significantly reduced gene expression (Figure 5c,d).

The release of IL-2 was determined using enzyme immunoassay. PBMC was incubated with HMGB1 for 6 days. An aliquot of the conditioned medium was selected every day to determine the concentration of IL-2. The amount of IL-2 in the conditioned environment of PBMC incubated with HMGB1 increases and reaches a peak on the fourth day. Preincubation of cells with TREM-1 LP17 inhibitor significantly inhibited the release of IL-2 (Figure 5b).

Thus, the interaction of HMGB1 with TREM-1 induces the expression of pro-inflammatory cytokines, as well as cytokines involved in the activation of cytotoxic lymphocytes.

### 2.5. HMGB1 Stimulates the Cytotoxic Activity of NK Cells and CD4+ and CD8+-T Lymphocytes

Next, populations of cytotoxic lymphocytes activated by HMGB1 were characterized. The highest activity was observed on the fourth and sixth days of incubation of the ligand with cells (Figure 4a). Cytotoxic effect on tumor cells of three populations of lymphocytes was studied: NK cells, CD4+ and CD8+T lymphocytes.

Subpopulations of NK cells, CD4+ and CD8+ T lymphocytes were isolated from PBMC by magnetic separation and their cytotoxic activity was determined (Figure 6a). It can be seen that on the fourth day of incubation, NK cells and CD4+-T lymphocytes showed cytotoxic activity, but not CD8+-T lymphocytes. On the sixth day, CD4+-T lymphocytes and CD8+-T lymphocytes had cytotoxicity, and NK cells were inactive (Figure 6b).

### 2.6. Activated by HMGB1 NK Cells and Cytotoxic T Lymphocytes Induce Various Mechanisms of Programmed Cell Death in Tumor Cells

Further, the mechanisms of programmed cell death under the action of HMGB1-induced cytotoxic lymphocytes were investigated. The results are shown in Figure 7.

It is known that cytotoxic processes of programmed cell death, apoptosis and necroptosis, develop at different rates. Therefore, cytotoxic lymphocytes were incubated with K562 tumor cells for 3 and 20 h.

It can be seen that NK cells exhibit cytotoxic activity only on the fourth day and only after 3 h of incubation with target cells. Tumor cells were recognized by NK cells via the interaction of a specific NKG2D receptor with a non-canonical HLA MicA molecule on the surface of target cells. NK cells secreted granzymes into the cell contact zone and induced caspase-dependent apoptosis in tumor cells. Antibodies against NKG2D, MicA and granzyme B, as well as inhibitors of caspase 3 and caspase 8 completely blocked cytotoxic activity (Figure 7a).

CD8+-T lymphocytes showed cytotoxic activity only on the sixth day after 3 h and after 20 h of incubation with target cells. These lymphocytes recognize tumor cells via the interaction of the NKG2D receptor present on the surface of CD8+-T lymphocytes, as it was shown for NK cells, with MicA target cells. However, CD8+-T lymphocytes kill tumor cells via FasL-Fas interaction. Antibodies against NKG2D, MicA, FasL and Fas, as well as inhibitors of caspase-3 and -8 and RIP1 kinase (necrostatin) inhibit cytotoxic activity (Figure 7b).

Cytotoxic CD4+-T lymphocytes kill tumor cells both on the fourth and sixth days after 3 h and after 20 h of incubation with tumor cells via FasL-Fas interaction, causing caspase-dependent apoptosis or RIP1 kinase-dependent necroptosis in them. Target cells are recognized via the interaction of the Tag7 CD4+-T lymphocyte with the main heat shock protein Hsp70 on the surface of tumor cells (Figure 7c).

Thus, HMGB1 induces the appearance of active NK cells, as well as non-canonical CD4+ and CD8+-T lymphocytes, killing HLA-negative tumor cells that have escaped immune control.

## 3. Discussion

The results presented here allow us to draw two important conclusions: (1) HMGB1 induces activation of the TREM-1 pathway; (2) HMGB1 promotes activation of non-canonical cytotoxic T-lymphocytes capable of lysing HLA-negative tumor cells.

HMGB1 was first described in 1973 as a structural chromatin protein. It prevents DNA damage by binding to both DNA and histones, it supports the structure of nucleosomes and it affects the packaging of DNA in chromatin [23,24,25].

Recently, extracellular HMGB1 has attracted serious attention. HMGB1 can appear in the extracellular environment due to two well-studied mechanisms: active secretion and passive release. Extracellular HMGB1 acts as a DAMP (damage-associated molecular pattern) and participates in the regulation of inflammation and immune response. The regulatory action of HMGB1 can be carried out due to its interaction with proinflammatory receptors.

It is known that HMGB1 can bind on the cell surface to RAGE, TLR2, TLR4 and TLR9 receptors [26,27,28]. HMGB1 is also considered a potential ligand of the proinflammatory TREM-1 receptor. As mentioned above, using immunoprecipitation, it was found that HMGB1 interacts with TREM-1-Fc. This interaction was confirmed by the formation of a stable HMGB1-TREM-1 complex in the presence of a DMA crosslinking agent. The binding between HMGB1 and TREM-1 was studied using BIAcore; the resulting constant was 3.5 × 10^−5^ M, which indicated a low affinity of these proteins [22]. Necrotic cell lysates containing HMGB1 are capable of triggering cytokine production and causing an inflammatory response [22]. It has been demonstrated that in induced cerebral hemorrhage in mice, TREM-1 is involved in conducting an intracellular signal via the PKC δ/CARD9 pathway. HMGB1 has been shown to participate in signal transmission via the TREM-1 receptor [29]. However, no direct interaction of HMGB1 with TREM-1 on the cell surface has been shown. Moreover, it is suggested that HMGB1 alone is not capable of triggering TREM-1 activation without coactivating molecules.

Here we have demonstrated the high affinity of the HMGB1-TREM-1 complex. The interaction of these proteins was quantified using microscale thermophoresis. The calculated K_d_ was 3 nM, which indicated the high affinity of these molecules and the stability of the studied complex.

The use of the BS^3^ crosslinking reagent and electrophoresis in SDS-PAGE followed by a Western blot with specific antibodies allowed us to show the binding of HMGB1 to TREM-1 on the cell surface of monocytes. HMGB1 binds to the inactive monomer TREM-1. These results were confirmed by confocal microscopy. It was possible to visualize the colocalization of HMGB1 with TREM-1, indicating the interaction of these proteins.

The detection of sTREM-1, recognized as a biomarker of TREM-1 pathway activation, in the conditioned medium of PBMC incubated with HMGB1 indicates that HMGB1 can be considered an activating ligand of TREM-1. The amount of sTREM-1 in the conditioned medium increased proportionally with an increase in the concentration of purified HMGB1. Consequently, HMGB1 binds to the receptor and triggers its activation without the participation of coactivating molecules.

We also demonstrated a new function of HMGB1—its ability to activate cytotoxic lymphocytes. In the modern literature, the activation of TREM-1 is mainly associated with the expression of cytokine/chemokine genes and the regulation of the inflammatory immune response. However, we have previously shown that the interaction of TREM-1 with its ligands is essential for the regulation of antitumor immunity. It has been shown that the TREM-1 ligands Tag7 and Hsp70 increase the expression of cytokine genes and also activate NK cells, CD4+ and CD8+ T lymphocytes [19,21]. Tag7-induced secretion of TNF and Ifnγ cytokines by CD14+ monocytes has been shown to lead to the induction of IL2 gene expression in CD4+T lymphocytes and subsequent secretion of this cytokine [19]. IL-2 is known to play a key role in the activation of cytotoxic lymphocytes.

Previously, we investigated the effect of IL-2 on the activation of cytotoxic lymphocytes and for the first time showed that CD4+ and CD8+ cytotoxic T lymphocytes are generated under the action of this cytokine, killing HLA-negative tumor cells via FasL-Fas interaction [30,31].

Here we have shown that HMGB1, as well as Tag7 or Hsp70, causes the expression of proinflammatory cytokine and Ifnγ genes and the release of IL-2 into the conditioned medium. The mRNA level of these cytokines decreased significantly in the presence of the LP17 inhibitor, which indicated the involvement of TREM-1 in this process. At the same time, gene expression was not completely suppressed, suggesting the interaction of HMGB1 with several receptors. It is known that both TLRs and RAGE are represented on monocytes, for which HMGB1 is also a ligand.

HMGB1 also induced the ability of PBMC to cause tumor cell death. In this case, the functional activity of this ligand was completely suppressed with the addition of the TREM-1 inhibitor, allowing the exclusion of the participation of other receptors in the transmission of the activation signal for cytotoxic cells.

HMGB1 induced activation of the same populations of cytotoxic lymphocytes that appeared after the interaction of TREM-1 with Tag7 or Hsp70: NK cells, CD4+ and CD8+ T lymphocytes. NK cells caused the death of tumor cells due to the secretion of granzymes into the contact zone. The cytotoxic effect of CD4+ and CD8+ T-lymphocytes was caused by the induction of apoptotic and necroptotic processes in tumor cells after the interaction of the FasL lymphocyte with the Fas receptor of the tumor cell.

Thus, we confirmed that HMGB1 is a TREM-1 ligand and showed that it can activate cytotoxic lymphocytes that kill tumor cells that evade immune control. Expanding the spectra of proinflammatory receptor ligands and understanding the mechanisms of their action is essential for the creation of new immunotherapy pathways.

## 4. Materials and Methods

### 4.1. Cell Culture and Sorting

Human erythroblastoma K562 cells were cultured in RPMI-1640 (Himedia Laboratories Private Limited, Maharashtra, India) and 10% FCS (Cytiva Livescience™, Marlborough, MA, USA). This cell line was obtained from the cell line collection of N. N. Blokhin National Medical Research Center of Oncology of the Ministry of Health of Russia. Human peripheral blood mononuclear cells (PBMC) were isolated from the total leukocyte pool of healthy donors by centrifugation in a Ficoll-Paque density gradient (Cytiva Livescience™, Marlborough, MA, USA), and cultured at a density of 4 × 10^6^ cells/mL in RPMI-1640 with 10^−9^ M HMGB1, BSA (10^−9^ M) or Tag7 in a concentration of 10^−9^ M for a certain time. All procedures performed were in accordance with the Declaration of Helsinki (1964) and its later amendments (World Medical Association, 2013) or comparable ethical standards and were approved by the medical ethics committee of FSBI N.N. Blokhin National Medical Research Center of Oncology of the Ministry of Health of the Russian Federation. The cells were sorted using sets of magnetic beads for isolation CD14+ (Thermo Fisher Scientific, Waltham, MA, USA), CD16+ CD56+ (Thermo Fisher Scientific, Waltham, MA, USA), CD8+ (Thermo Fisher Scientific, Waltham, MA, USA) and CD4+ (Thermo Fisher Scientific, Waltham, MA, USA) according to the manufacturer’s protocol.

### 4.2. Proteins and Antibodies

HMGB1 was obtained as described in [30]. Pierce™ Chromogenic Endotoxin Quant Kit (Thermo Fisher Scientific, Rockford, Illinois, USA) was used to confirm absence of LPS in recombinant protein. Tag7 was obtained as described [31], sTREM-1 was obtained according to [30]. The BSA was from Sigma (Sigma–Aldrich, St. Louis, MI, USA). The inhibitory peptide LP17 (LQVTDSGLYRCVIYHPP, 10^−9^ M) was preincubated with lymphocytes for 1 h, and then HMGB1 was added. Polyclonal antibodies to Tag7 (PGLYRP1; RRID:AB_10960204) were obtained from ELK Biotechnology (ELK Biotechnology, Denver, CO, USA), antibodies to MicA (RRID:AB_10607089) were obtained from Sigma (Sigma–Aldrich, St. Louis, Missouri, USA); antibodies to Hsp70 (AB_2866870) were from Invitrogen (Thermo Fisher Scientific, Waltham, MA, USA); antibodies to Granzyme B (RRID:AB_304251) and TREM-1 (RRID:AB_10864342) were obtained from ABclonal (ABclonal, Woburn, MA, USA); antibodies to NKG2D (RRID:AB_628024), the C-end of FasL (C-178; RRID:AB_2246664) were obtained from Santa Cruz Biotechnology (Santa Cruz Biotechnology, Santa Cruz, CA, USA) and Fas antibody (C-20) (Santa Cruz Biotechnology, Dallas, TX, USA).

### 4.3. Affinity Chromatography, Immunoadsorption and Immunoblotting

Monocytes (5 × 10^7^ cells) isolated from the total PBMC pool were incubated with HMGB1 (10^−8^ M) in the presence of BS^3^ (Thermo Fisher Scientific, Waltham, MA, USA), then lysed in RIPA buffer (Sigma–Aldrich, St. Louis, MO, USA) and purified with using Dynabeads M-280 Sheep Anti-Rabbit IgG (Thermo Fisher Scientific, Waltham, MA, USA) conjugated with antibodies against TREM-1, according to the manufacturer’s protocol. Further, this fraction was resolved using 12% SDS-PAGE followed by Western blotting. Detected using primary antibodies to HMGB1 (1:1000, 1 h) (ABclonal, Woburn, MA, USA) and secondary antibodies ECL™ anti-Rabbit IgG, peroxidase-linked species-specific whole antibody (from donkey) secondary antibody (Cytiva Livescience™, Marlborough, MA, USA) (1:10,000; 1 h). The results were visualized using the ECL Plus kit (GE Healthcare, Chicago, IL, USA) in accordance with the manufacturer’s protocol. Chemiluminescence was recorded using iBright (Thermo Fisher Scientific, Boston, MA, USA).

### 4.4. Cytotoxicity Assays

Cells of the K562 line were cultured in 96-well plates (Corning World Headquarters, Corning, NY, USA) (6 × 10^4^ per well), then lymphocytes were added in a ratio of 20:1 and incubated at 37 °C in an atmosphere with 5% CO_2_ for 3–20 h. The inhibition test was performed using polyclonal antibodies (anti-NKG2D, anti-MicA, anti-Hsp70, anti-FasL and anti-granzyme B) at a concentration of 20 micrograms/mL. Cytotoxic activity of lymphocytes was determined using the Cytotox 96 Assay kit (Promega, Madison, WI, USA) according to the manufacturer’s protocol. In inhibition assays, cells were pre-incubated for 1 h with a caspase 3 inhibitor Ac-DEVD-CHO (Selleck Chemicals LLC, Houston, TX, USA), a caspase 8 inhibitor Ac-IETD-CHO (Selleck Chemicals LLC, Houston, TX, USA) at a concentration of 50 mkM each or with a RIP1 kinase inhibitor necrostatin 1 (C = 5 mM) (Sigma–Aldrich, St. Louis, MO, USA), then lymphocytes were added.

### 4.5. IFA

The level of IL2 secretion was tested using the ELISA Human IL-2 kit (Bender MedSystems GmbH, Vienna, Austria) in accordance with the manufacturer’s protocols. The level of sTREM-1 secretion was studied using the TREM-1 Human ELISA kit (Life Technologies Corporation, Carlsbad, CA, USA).

### 4.6. RT-PCR

RNA was isolated from the first and second PBMC fractions after their treatment with HMGB1 (10^−9^ M) for 3 h. The second PBMC fraction was preincubated with LP17 for 1 h. PBMC was lysed in 500 mL of the Reagent “RNA Extract” (“Eurogen”, Russia) according to the manufacturer’s protocol. The RNA was measured using a NanoDrop device (Thermo Fisher Scientific, Waltham, MA, USA) and then equal amounts of RNA (1.1 micrograms) were used. Electrophoresis was used to detect RNA degradation. cDNA synthesis was carried out using oligo (dT) primers (Eurogen, Russia). The products were used for cPCR with primers for genes encoding RPLP0, TNFa, IL1ß, IL6 and IFNy. The RPLP0 mRNA level was taken as a reference. The primers were as follows: for RPLP0: 5′ACTGGAGACAAAGTGGGAGCC 3′(forward) 5′ CAGACACTGGCAACATTGCG 3′ (reverse), for IFNy: 5′ GGGTTCTCTTGGCTGTTACTG 3′ (forward), 5′ TTCGTCACTCTCCTCTTCCA 3′ (reverse); for TNFa 5′-CTCTTCTGCCTGCTGCACTTTG-3′ (straight), 5′-ATGGGCTACAGGCTTGTCACTC-3′ (reverse); IL1ß 5′-CCACAGACCTTCCAGGAGAATG-3′ (forward) 5′-GTGCAGTTCAGTGATCGTACAGG-3′(reverse); IL6 5′-AGACAGCCACTCACCTCTTCAG-3′ (direct) 5′-TTCTGCCAGTGCCTCTTTGCTG-3′(reverse).

The measurements were carried out in at least three repetitions and the average value was calculated. Expression levels were quantified using the 2ΔΔCt method.

### 4.7. Microscale Thermophoresis

A fluorescent label was added to the purified TREM-1 protein using the Alexa Fluor™ 633 Protein Labeling Kit (Life Technologies Corporation, Eugene, OR, USA) in accordance with the manufacturer’s protocol. HMGB1 and TREM-1 (200 nM) were incubated for 20 min in the dark at room temperature in 16 different concentrations obtained by sequential dilution, starting with the highest soluble concentration. The samples were transferred to glass capillaries (Monolith NT Capillaries) and analyzed by microscale thermophoresis on a Nano-Temperature Monolith NT 115 device (20% IR laser power). The signal quality was monitored using a NanoTemper Monolith device to detect possible autofluorescence of the ligand, aggregation, or changes in the rate of photobleaching. The experiments were carried out in at least three repetitions and processed using affinity analysis software (MO Control v.1.6.1, NanoTemper Technologies GmbH, München, Germany).

### 4.8. Confocal Microscopy

CD14+ cells were isolated from the total PBMC pool by magnetic separation (as indicated earlier). HMGB1 protein was added to them at a concentration of 5 × 10^−8^ M, then the cells were washed with PBS twice, then fixed with a cold solution of 4% formaldehyde. The TREM-1 receptor was stained with monoclonal mouse antibodies against TREM-1 and Goat anti-mouse IgG (H + L) Cross-Adsorbed Secondary Antibody, Alexa Fluor™ 488 (Molecular Probes By Life Technologies, Carlsbad, CA, USA), and for HMGB1–polyclonal rabbit antibodies against HMGB1 and Goat anti-Rabbit IgG (H + L) Cross-Adsorbed Secondary Antibody, Alexa Fluor™ 633 (Molecular Probes By Life Technologies, Carlsbad, CA, USA). Fluorescence images were obtained using a Leica STELLARIS 5 confocal microscope (Leica, Wetzlar, Germany), analyzed using Leica confocal software (2.61.15) and processed in Photoshop CE (Adobe Systems, San Jose, CA, USA).

### 4.9. Statistical Analysis

All the data given in the article were calculated for at least three independent repeats. The data is presented as an average ± standard deviation. Testing for significant differences between treatment and control was carried out using MathCad software (version 15.0, PTC, Cambridge, Great Britain) in experiments on the treatment of cells with one agent, Student criteria were used; for experiments on the treatment of cells with two agents or more, two-factor analysis of variance was used.

## 5. Conclusions

In this work, we have found that HMGB1 induces activation of the TREM-1 pathway. This HMGB1-induced activation leads to the appearance in PBMC of non-canonical cytotoxic T-lymphocytes capable of lysing HLA-negative tumor cells. Expanding the spectra of proinflammatory receptor ligands and understanding the mechanisms of their action is essential for the creation of new immunotherapy pathways.

## Figures and Tables

**Figure 1 ijms-25-00627-f001:**
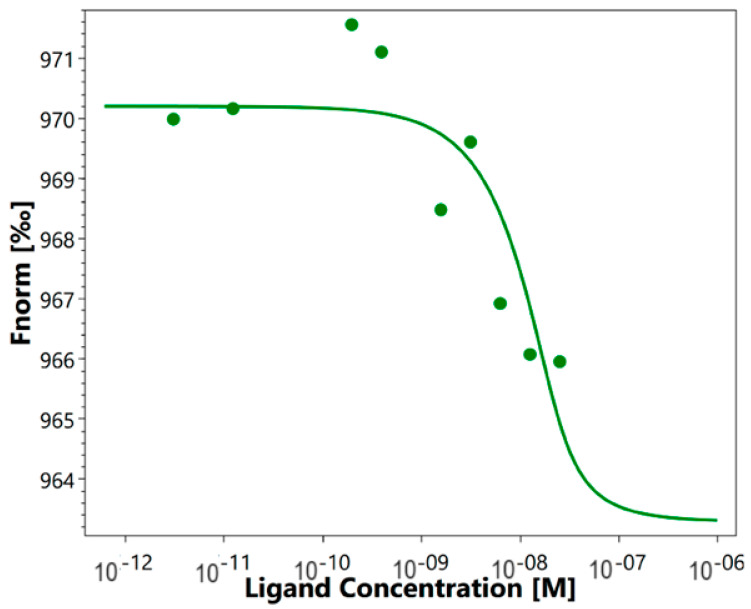
Microscale thermophoresis data for the interaction of labeled by the Alexa Fluor™ 633 Protein Labeling Kit sTREM-1 with HMGB1. Experiment was carried out in triplicate, and the most common data are shown.

**Figure 2 ijms-25-00627-f002:**
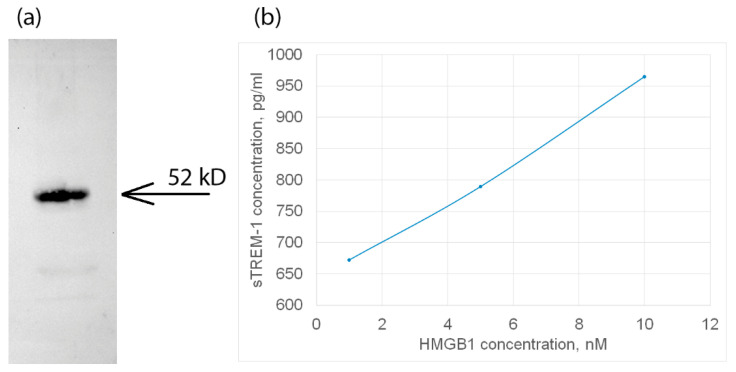
(**a**) The HMGB1 protein was applied to sTREM-1 conjugated with Sepharose, eluted with acetonitrile, and resolved using SDS PAGE and Western blot with antibodies to HMGB1. (**b**) ELISA of soluble TREM-1 in lymphocytes in presence of different concentrations of HMGB1 protein. The cultured medium was collected after 24 h of incubation with various concentrations of HMGB1. Experiment was carried out in triplicate, and the most common data are shown.

**Figure 3 ijms-25-00627-f003:**
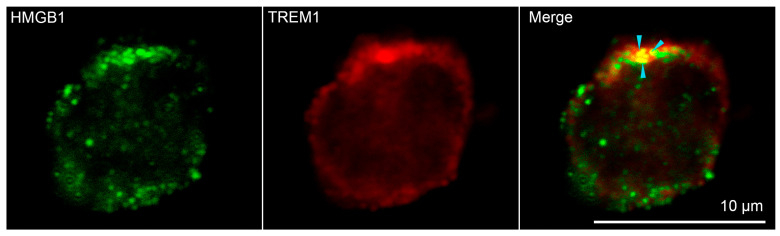
Confocal micrograph of HMGB1 (green) and TREM-1 (red) and layers superposition on the surface of monocyte. Arrows indicate colocalization region. About 50% of all observed cells were double stained. (n = 4) For analysis ImageJ Software (V 1.8.0) was used.

**Figure 4 ijms-25-00627-f004:**
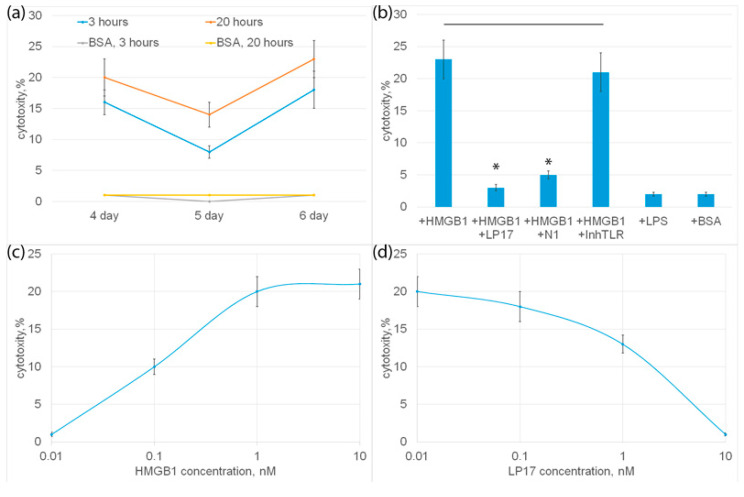
(**a**) Cytotoxic activity of PBMC after 4, 5 or 6 day incubation with HMGB1 protein, calculated after 3 or 20 h incubation with K562 cells. As a control, cytotoxic activity of PBMC without HMGB1 added is presented. (**b**) Cytotoxic activity of PBMC after 6-day incubation with HMGB1 protein in presence of LP17 peptide, N1 peptide or TLR inhibitor, calculated after 20 h incubation with K562 cells. Cytotoxic activity of PBMC incubated with BSA orLPS is presented as a control. (**c**) Cytotoxic activity of PBMC after 6-day incubation with various concentrations of HMGB1 protein. Cells were collected after 6 days and than added to K562 cells for 20 h and cytotoxic activity was calculated. (**d**) Cytotoxic activity of PBMC after 6-day incubation with HMGB1 protein in presence of various concentrations of LP17 peptide, calculated after 20 h incubation with K562 cells. n = 5 for each groups). (*p*-value: * < 0.05).

**Figure 5 ijms-25-00627-f005:**
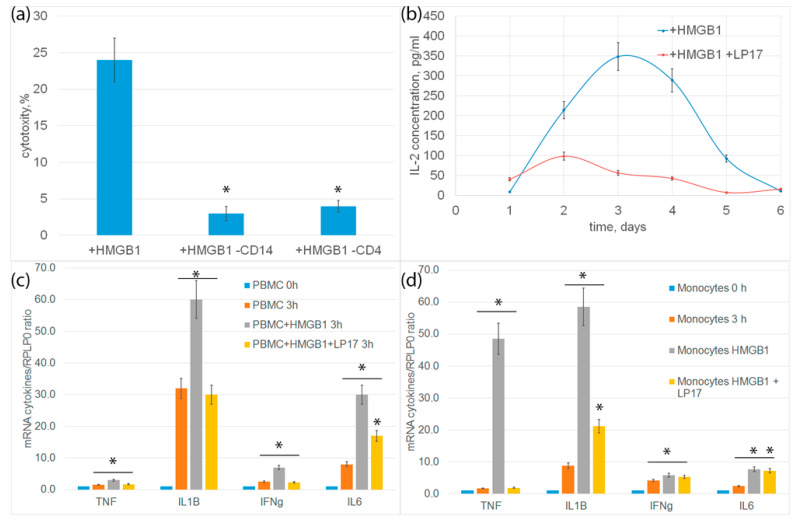
(**a**) Cytotoxic activity of PBMC or PBMC deprived from monocytes or CD4+ T cells after 6-day incubation with HMGB1 protein, calculated after 20 h incubation with K562 cells. (**b**) ELISE of secreted IL-2 cytokine concentration dependence on time of incubation with HMGB1 protein. (**c**) rtPCR analysis of cytokines IL-1b, IL-6, TNF and IFH-g in PBMC or in isolated via magnetic separation subpopulation of monocytes (**d**) after addition of HMGB1 protein or HMGB1 protein +LP17 peptide. (n = 3 for each group). (*p*-value: * < 0.05).

**Figure 6 ijms-25-00627-f006:**
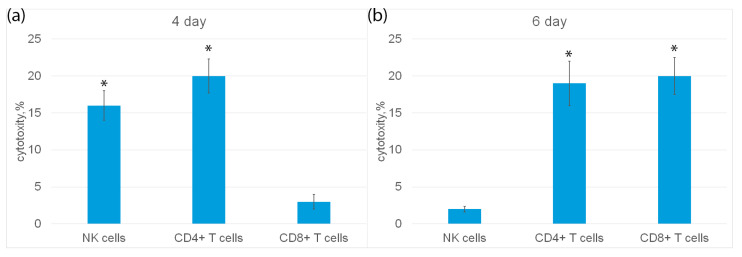
Cytotoxic activity of subpopulations of PBMC, purified on day 4 (**a**) or day 6 (**b**) using magnetic separation and added to K562 cells for 3 h. n = 5 for each group). (*p*-value: * < 0.05).

**Figure 7 ijms-25-00627-f007:**
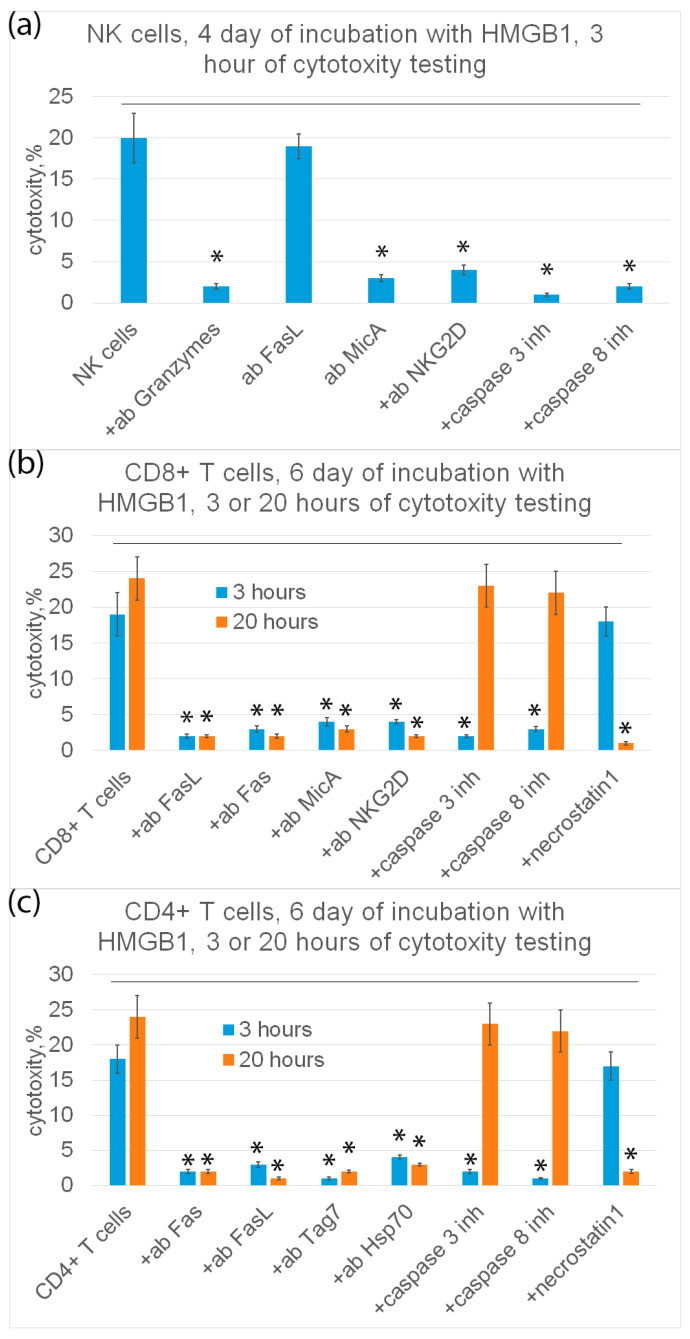
Cytotoxic activity of subpopulations of PBMC (NK cells (**a**), CD8+ T cells (**b**) or CD4+ T cells (**c**)) purified on day 4 (**a**) or day 6 (**b**,**c**) and added to K562 cells for 3 (**a**–**c**) or 20 (**b**,**c**) hours in presence of various inhibitory antibodies or inhibitors. n = 5 for each group. (*p*-value: * < 0.05).

## Data Availability

Data is contained within the article and Appendix A.

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
