# Peer review of "The Interaction of HMGB1 with the Proinflammatory TREM-1 Receptor Generates Cytotoxic Lymphocytes Active against HLA-Negative Tumor Cells"

_ijms, 2024, doi:10.3390/ijms25010627_

Round 1

Reviewer 1 Report

Comments and Suggestions for Authors

This manuscript by Yurkina et al, aimed at investigating whether HMGB1-TREM-1 interaction exists in PBMC-derived cells and is able to produce an effective cytotoxic response against HLA-negative tumor cells. The authors demonstrate that HMGB1 protein activates the TREM-1 pathway, and consequently the production of several cytokines, including IL-2, IL-1b, IL-6, TNF or IFNy. Furthermore, the authors showed that the above mentioned interaction are strongly correlated with the immune response against HLA-negative tumor cells  mainly triggered by non-canonical cytotoxic T-lymphocytes. The article is well structured, the reading is clear and easy to follow and the results obtained are useful and suitable for further studies to get a more in-depth study of this novel activation pathway .After having carefully reviewed the manuscript as well as figures and supplemental data, I recommend the article for publication after solve few questions: 

Major comments:

- For detecting direct HMGB1-TREM-1 after BS3 treatment, a  mock control with unspecific IgG should be included.

- Regarding confocal microscopy and co-localization experiments, no statistical data was included. The unique image shown is not clear enough, and quantification and statistical data must be incorporated.

- Results in section 2.3 are not supported by controls: measurements without HMGB1 supplementation should be included.

- In section 2.4.: Why cytokine measurements are done from PBMCs and not directly from CD14+ monocytes and CD4+ T cells previously isolated?

Minor comments:

Abstract.- Line 11. Delete “the” in “the cells of damaged tissues.

1.- Lines 85-86. Sentence not clear. What it means “under a control”?

1.- Lines 92-93. HMGB1 is not only nuclear, but also cytoplasmatic.

2.- Line 110. Standard errors should be included in Kd calculations.

2.- Lines 178-181. Standard errors should be included in IC50  calculations.

2.- Line 216. Include a comma between “cells” and “CD4+”

Figures. There should be homogeneity in terms of the format of the figures. They seem to have been taken from different statistical packages. The fonts are also different.

Author Response

Major comments:

- For detecting direct HMGB1-TREM-1 after BS3 treatment, a  mock control with unspecific IgG should be included.

We are thankful to the reviewer for careful reading of our work and helpful suggestions. We have added several control panels in the Supplemental Materials to address this comment.

- Regarding confocal microscopy and co-localization experiments, no statistical data was included. The unique image shown is not clear enough, and quantification and statistical data must be incorporated.

We have done additional experiments to address this concern and have added statistical data and several additional panels in Supplemental Material Section. 50% of cells with added HMGB1 show colocalization with TREM-1 receptor. In this work, we use confocal microscopy to demonstrate that HMGB1 is able to bind to TREM-1 on the cell surface. For the preparation of extensive statistical data, we need more time, that it was given to us by Editors to complete the submission of our manuscript.

- Results in section 2.3 are not supported by controls: measurements without HMGB1 supplementation should be included.

We have added control data with no HMGB1 added to the Figure 4a to improve quality of data provided in section 2.3 of our manuscript.

- In section 2.4.: Why cytokine measurements are done from PBMCs and not directly from CD14+ monocytes and CD4+ T cells previously isolated?

We have done additional PCR experiment on isolated populations of monocytes and CD4+ T cells. Monocytes subpopulation has TREM-1 on the surface and provide activation signal. We have added this data to the figure 5. CD4+ subpopulation is activated later induced by cytokines released by monocytes. At selected time point CD4+ cells are still inactive. We have added this data to the Supplemental Section.

Minor comments:

 Abstract.- Line 11. Delete “the” in “the cells of damaged tissues.

 1.- Lines 85-86. Sentence not clear. What it means “under a control”?

 1.- Lines 92-93. HMGB1 is not only nuclear, but also cytoplasmatic.

 2.- Line 110. Standard errors should be included in Kd calculations.

 2.- Lines 178-181. Standard errors should be included in IC50  calculations.

 2.- Line 216. Include a comma between “cells” and “CD4+”

 Figures. There should be homogeneity in terms of the format of the figures. They seem to have been taken from different statistical packages. The fonts are also different.

We are thankful to the reviewer for careful reading of our manuscript and uncovered flaws. We have made all listed corrections to the text to improve its quality.

Reviewer 2 Report

Comments and Suggestions for Authors

This manuscript raises concerns regarding the claim that HMGB1 activates the TREM-1 pathway due to weak evidence. There are several issues that need to be addressed, particularly in terms of experimental design and suitable controls. Some specific concerns include:

  1. In Figure 2a, it is important to understand the situation without the crosslinking agent and to assess the specificity of the antibodies used for the pull-down experiments.

  2. Regarding Figure 4a, it would be helpful to determine the percentage of cytotoxicity without any treatment (i.e., no HMGB1 protein added). It seems that there is no significant difference between the two groups and across different time points.

  3. In Figure 4b, it is unclear why the treatment of LPS resulted in the same percentage of cytotoxicity as the treatment with BSA. This raises questions about the specificity and potential confounding factors.

  4. In Figure 5d, why without treatment with HMGB1, the expression of the genes for group of PBMC 3h increase so much as compare to PBMC 0h? 
  5. The inconsistent effects of LP17 treatment in inhibiting inflammation and cytotoxicity are indeed concerning. It is worth noting that Figure 4b shows a significant effect, while Figure 5b demonstrates no change at all. If the discrepancy is due to dosage and different time points, it would be advisable to adjust these variables in order to achieve consistent results. Ensuring consistency across experiments will strengthen the reliability and robustness of the findings.

Author Response

  1. In Figure 2a, it is important to understand the situation without the crosslinking agent and to assess the specificity of the antibodies used for the pull-down experiments.

We thank the reviewer for careful reading of our work. We have added control panels in the Supplemental Material to support our data.

  1. Regarding Figure 4a, it would be helpful to determine the percentage of cytotoxicity without any treatment (i.e., no HMGB1 protein added). It seems that there is no significant difference between the two groups and across different time points.

We have added additional control data on the Figure 4a, showing absence of cytotoxic activity when HMGB1 protein was not added. Cytotoxic activity was calculated from the same lymphocytes on different time point (red line – 3 hours – apoptosis, and blue line – 20 hours –necroptosis). Indeed the difference between percentage of cells dying via apoptosis and necroptosis is not big. This graph is presented to support our choice of the best time point to collect activated lymphocytes and length of incubation with target cells to measure cytotoxic effect of induced lymphocytes. Also it helps us to conclude that both apoptotic and necroptotic processes are induced in target cancer cells.

  1. In Figure 4b, it is unclear why the treatment of LPS resulted in the same percentage of cytotoxicity as the treatment with BSA. This raises questions about the specificity and potential confounding factors.

Both LPS treatment and addition of BSA protein did not cause appearance of cytotoxic lymphocytes, able to kill HLA-negative cancer cells K562. Statistical analysis shows no difference between cytotoxity 1-2% in BSA or LPS induced lymphocytes and spontaneous control cell death.

  1. In Figure 5d, why without treatment with HMGB1, the expression of the genes for group of PBMC 3h increase so much as compare to PBMC 0h? 

In our work we use the blood of different healthy donors with unknown background. We have done several additional experiment to collect better statistical data and presented its results in the Supplemental Material Section. In several control experiments we also have noticed activation of some cytokines without HMGB1 treatment. We think that this activation depends on immune status of selected donor.

  1. The inconsistent effects of LP17 treatment in inhibiting inflammation and cytotoxicity are indeed concerning. It is worth noting that Figure 4b shows a significant effect, while Figure 5b demonstrates no change at all. If the discrepancy is due to dosage and different time points, it would be advisable to adjust these variables in order to achieve consistent results. Ensuring consistency across experiments will strengthen the reliability and robustness of the findings.

We thank the reviewer for careful reading of our work and helpful questions. We have performed additional experiments and also have added the data of PCR analysis of cytokines, induced in monocytes population. Also we have changed axis on the figure 5d from exponential one to the linear one to show that addition of HMGB1 provides more than two times increase in production of cytokines mRNA above unactivated for 3 hours PBMC cells, and this increase is statistically significant. Moreover, addition of LP17 inhibitory peptide decreases mRNA level activated cells to the level of unactivated for 3 hours PBMC cells, almost fully inhibiting the effect of HMGB1 addition. We believe that in both cases in Figure 4 and in Figure 5 LP17 does the same thing, reducing the effect of HMGB1 activation almost to the control level.

Round 2

Reviewer 1 Report

Comments and Suggestions for Authors

All issues were correctly solved. No extra suggestions.

Author Response

We thank the reviewer for interesting questions and careful reading of our work.

Reviewer 2 Report

Comments and Suggestions for Authors

The authors have not adequately addressed my concerns. I would like to reiterate the following points:

  1. The figure titles and the unit (mkg) used in Supplemental Figures 1 and 2 are unusual. According to Google, mkg is the SI unit for meter-kilogram. Additionally, I am curious about the presence of a band in lane 4 of Supplemental Figure 1 when no crosslinking agent was mentioned in the methodology.

  2. In Figure 4A, two curves are presented without clear explanation or identification. The authors have not provided a satisfactory explanation for why the treatment for 3 hours exhibits higher cytotoxicity compared to the 20-hour treatment.

  3. The authors have not sufficiently explained the inclusion of LPS (lipopolysaccharide) and BSA (bovine serum albumin) as controls in Figure 4B. Are the proteins used in this study contaminated with LPS? If so, was the level of LPS contamination comparable to the concentration used in the control group?

  4. Supplemental Figure 5 lacks proper labeling and units on the y-axis, and the figure itself is poorly drawn. The legend accompanying the figure is also inadequately written. It remains unclear what the difference is between the four groups and the left and right sides of the figure. Additionally, the authors' explanation in their response has not alleviated my concern that all observations may be merely correlational, suggesting that the treatment of HMGB1 is not critical.

Comments on the Quality of English Language

Pls refer to my comments in above section. 

Author Response

The authors have not adequately addressed my concerns. I would like to reiterate the following points:

1. The figure titles and the unit (mkg) used in Supplemental Figures 1 and 2 are unusual. According to Google, mkg is the SI unit for meter-kilogram. Additionally, I am curious about the presence of a band in lane 4 of Supplemental Figure 1 when no crosslinking agent was mentioned in the methodology.

We are grateful to the reviewer for useful comments. We have changed “mkg” to the “µg”. We have repeated the experiment showed in Supplemental Figure 1 and have earned the same results. Indeed complex HMGB1-TREM-1 is stable enough to retain under the experimental conditions and dissociates only after addition of SDS and mercaptoethanol during SDS PAGE probes preparation. Dissociated from complex HMGB1 is stained by specific antibodies and is visible in lane 4 on Supplemental Figure 1.

2. In Figure 4A, two curves are presented without clear explanation or identification. The authors have not provided a satisfactory explanation for why the treatment for 3 hours exhibits higher cytotoxicity compared to the 20-hour treatment.

We thank the reviewer for careful reading of our work. In the Figure 4A we have incorrectly designated 3 hours cytotoxity and 20 hours cytotoxity. We have corrected this issue and changed Figure 4A. 20 hours cytotoxity is higher than 3 hour cytotoxity.

3. The authors have not sufficiently explained the inclusion of LPS (lipopolysaccharide) and BSA (bovine serum albumin) as controls in Figure 4B. Are the proteins used in this study contaminated with LPS? If so, was the level of LPS contamination comparable to the concentration used in the control group?

In our work we are trying to postulate that HMGB1 is activating lymphocytes via TREM-1 receptor. To provide the adequate controls for our conclusions we need protein, that did not interact with TREM-1 receptor. We have shown recently that BSA did not interact with TREM-1 receptor (Sharapova et al. 2022) and we take BSA for control experiment. It was proposed by several authors (24 ref. in manuscript) that HMGB1 is able to interact with Toll receptors. To provide controls that Toll receptors activation is not needed for generation of cytotoxic lymphocytes in our system we need Toll receptor activator. It is known that LPS is the specific activator of Toll receptors. We use LPS to show that Toll receptor activation do not lead to the activation of cytotoxic lymphocytes in our system. We have additionally described our reasons to choose BSA and LPS as controls in the text of our work to make it more understandable to the readers. For our recombinant HMGB1 protein we have tested the level of contamination by LPS and have not detected it, as we have reported in Materials and Methods Section of our manuscript.

4. Supplemental Figure 5 lacks proper labeling and units on the y-axis, and the figure itself is poorly drawn. The legend accompanying the figure is also inadequately written. It remains unclear what the difference is between the four groups and the left and right sides of the figure. Additionally, the authors' explanation in their response has not alleviated my concern that all observations may be merely correlational, suggesting that the treatment of HMGB1 is not critical.

We have changed Supplemental Figure 5 to show that the results of 4 different donors are presented. We have changed the legends to supplemental Figure 5 to provide more information to the readers of the manuscript. In all four cases we take lymphocytes from the four different donors and cultivate them for 3 hours in RPMI-1640 medium supplemented by glutamine without fetal calf serum addition. Cells on zero time point and after incubation for 3 hours were collected and used for preparation of RNA material for qPCR analysis. The results show that lymphocytes from different donors have different profile of cytokines activation and can show either reduction in level of selected cytokines or increase in it. We think that this cells behavior is due to different processes that take place in different donors before the procedure of lymphocytes isolation. The difference in immunological background of tested lymphocytes do not allow us to make extensive statistic investigations on different donors. Presented in the work statistical data are collected using parallel tests on selected donors.

We have tested lymphocytes form different donors to make the conclusion that addition of HMBG1 activates the cytokine production. In all cases tested addition of HMGB1 for 3 hours provides remarkable increase of RNA level of proinflammatory cytokines IL1b, IL-6 and TNF over untreated for 3 hours cells. Also we have added Figure 5D, where the data on isolated population of monocytes are presented. In this case as well as in total PBMC the RNA level of proinflammatory cytokines is strongly increased after HMGB1 addition.